# Guideline Concordance of Statin Treatment Decisions: A Retrospective Cohort Study

**DOI:** 10.3390/jcm9113719

**Published:** 2020-11-19

**Authors:** Yael Rachamin, Stefan Markun, Thomas Grischott, Thomas Rosemann, Rahel Meier

**Affiliations:** Institute of Primary Care, University of Zurich and University Hospital Zurich, Pestalozzistr. 24, 8091 Zurich, Switzerland; stefan.markun@usz.ch (S.M.); thomas.grischott@usz.ch (T.G.); thomas.rosemann@usz.ch (T.R.); rahel.meier@usz.ch (R.M.)

**Keywords:** guideline adherence, statins, cardiovascular diseases, low-density lipoprotein cholesterol, prevention

## Abstract

Guidelines recommend initiation of statins depending on cardiovascular risk and low-density lipoprotein cholesterol (LDL-C) levels. In this retrospective cohort study, we aimed to assess guideline concordance of statin treatment decisions and to find determinants of undertreatment in Swiss primary care in the period 2016–2019. We drew on electronic medical records of 8060 statin-naive patients (50.0% female, median age 59 years) with available LDL-C levels and cardiovascular risk. Guideline concordance was assessed based on the recommendations of the European Society of Cardiology, and multilevel logistic regression was performed to find determinants of undertreatment. We found that statin treatment was initiated in 10.2% of patients during one year of follow up. Treatment decisions were classified as guideline-concordant in 63.0%, as undertreatment in 35.8% and as overtreatment in 1.2%. Among determinants of undertreatment were small deviation from LDL-C treatment thresholds (odds ratio per decrease by 1 mmol/L: 2.09 [95% confidence interval 1.87–2.35]), high compared with very high cardiovascular risk (1.64 [1.30–2.05]), female sex (1.31 [1.05–1.64]), and being treated by older general practitioners (per 10 year decrease: 0.74 [0.61–0.90]). In conclusion, undertreatment of patients at high or very high cardiovascular risk was common, but general practitioners considered cardiovascular risk and LDL-C in their treatment decisions.

## 1. Introduction

Cardiovascular (CV) diseases are the major cause of death globally, responsible for 31% of worldwide deaths and 45% of deaths in Europe [1,2]. Increased blood lipid levels, particularly low-density lipoprotein cholesterol (LDL-C), are among the major risk factors for CV diseases [3]. Additional risk factors include elevated blood pressure, advanced patient age, and comorbidities such as diabetes, chronic kidney disease or established atherosclerotic CV disease [4]. Based on the occurrence and severity of these risk factors, the European Society of Cardiology (ESC) classifies patients into four risk categories, namely low, moderate, high, and very high risk [4].

To decrease the risk of fatal CV disease, guidelines suggest lowering LDL-C levels [4,5], and statins are the first-line therapy to achieve this goal [6]. Recommendations for the initiation of statins depend on CV risk and also directly on LDL-C levels because lowering LDL-C levels is most relevant for patients at increased CV risk [4].

Given the high prevalence of increased CV risk, guideline-concordant statin treatment is of great concern. Both statin undertreatment, i.e., forgoing statin treatment for patients for whom initiation is recommended, and overtreatment, i.e., the initiation of statin treatment in patients for whom it is not recommended, have been reported [7,8,9]. Undertreatment is of highest concern, as it signifies missed opportunities to effectively decrease CV mortality. To reduce undertreatment, knowledge of its prevalence and determinants is required.

Hence, with this study, we aimed to assess concordance of statin treatment decisions with guideline recommendations and to find determinants of undertreatment for patients in Swiss primary care.

## 2. Materials and Methods

### 2.1. Study Design and Setting

We performed a retrospective cohort study in Swiss primary care using data from the FIRE project (Family Medicine ICPC-Research using Electronic Medical Records) [10]. The FIRE project collects anonymized routine data from electronic medical records of contributing general practitioners (GPs). Since the project was started in 2009, more than 650 GPs have joined the FIRE project. At present, the database contains over nine million patient records with administrative information (patient age, gender), diagnosis codes according to International Classification of Primary Care 2nd edition (ICPC-2), laboratory and vital signs measures and drug treatment. Where the necessary information was available, the database has been augmented with the patients’ calculated CV risks according to the 2016 ESC/European Atherosclerosis society (EAS) guidelines for the management of dyslipidemias (ESC-G) [4,11].

The local Ethics Committee of the Canton of Zurich waived approval of the present study because the FIRE project was considered to lie outside the scope of the Human Research Act (BASEC-Nr. Req-2017-00797).

### 2.2. Patients

We included all patients with known CV risk and at least one LDL-C measurement in the period 01.09.2016–31.08.2018 to obtain a cohort with at least one year of follow up within the period of applicability of the ESC-G (01.09.2016–31.08.2019). For each patient, the first LDL-C measurement in the period 01.09.2016–31.08.2018 was set as the index measurement. Patients were excluded if they were treated with statins at, or prior to, the date of the index LDL-C measurement. Further, patients were excluded if they had no record in the FIRE database prior to one year before the index LDL-C measurement (baseline year) or no record at least one year after the index LDL-C measurement (follow up).The follow up period lasted until statin initiation, or, at the latest, one year after the index LDL-C measurement. The study phases are visualized in Figure 1A, and the selection process is shown in Figure 1B.

### 2.3. Data Query and Variables

We extracted data on the patients and the GPs they visited most often in their baseline year. Of GPs, we extracted sex and year of birth. Of patients, we extracted sex, year of birth, date of index LDL-C measurement and corresponding CV risk category according to the ESC-G (for our operationalization within the FIRE project, see Appendix A; for a detailed description, see Meier et al. [11]), date of statin initiation within one year of the index LDL-C measurement including initial treatment intensity (low, moderate, or high; calculated from product and daily dose according to the 2013 American College of Cardiology/American Heart Association guidelines [12]), date of diagnosis of relevant morbidities (atherosclerotic CV disease, diabetes, hypertension, moderate and severe chronic kidney disease, obesity; for definitions, see Appendix A). Within each patient’s follow up period, we assessed the mean LDL-C value and the date of the last LDL-C measurement. Additionally, we extracted data on consultations to calculate the continuity of care index (COCI) [13] during the observation period (start of baseline year to end of follow up).

### 2.4. Definitions of Treatment Recommendations and Guideline Concordance

Statin treatment recommendations were defined according to the ESC-G based on the patients’ CV risk categories and LDL-C values (Figure 2). The ESC-G distinguish three recommendations, which we termed “statin initiation”, “no statin”, and “statin optional”. The recommendation “statin initiation” pertains only to patients at very high or high CV risk; the treatment thresholds are 1.8 mmol/L and 2.6 mmol/L respectively. We considered the patients’ CV risks at the date of the index LDL-C measurements and their mean LDL-C values during follow up to identify the recommended interventions. We observed treatment decisions (statin initiation or non-initiation) for one year from the index LDL-C measurement to assess concordance with the guidelines. Undertreatment was defined as forgoing statin treatment for patients for whom treatment initiation was recommended, and overtreatment was defined as the initiation of statin treatment for patients for whom it was not recommended. For the recommendation “statin optional”, both treatment decisions were considered guideline-concordant.

### 2.5. Statistical Analysis

For data description, we used counts and proportions (*n* and %) or medians and interquartile ranges (IQR) as appropriate. To assess determinants of undertreatment, we built a multilevel logistic regression model with the practice and the GP within the practice as random effect variables. Preselected independent variables were patient age, gender, continuity of care, deviation of LDL-C levels from treatment thresholds, CV risk category, as well as GP age and gender. We used the R software (R Foundation for Statistical Computing, Vienna, Austria) [14] (Version 3.5.1) to perform the data analysis, and reported *p* values and 95% confidence intervals (95% CI). Missing data were left unchanged.

## 3. Results

We included 8060 patients (50.0% female) from 80 different practices in this study. Median patient age was 59 years (IQR = 51–68) and CV risk was low in 14.7% (*n* = 1184), moderate in 36.2% (*n* = 2919), high in 23.8% (*n* = 1918) and very high in 25.3% (*n* = 2039) of patients. Statin treatment was initiated in 10.2% of patients (*n* = 821) during follow up. For patients who were initiated on a statin, the median time interval from the index LDL-C measurement to statin initiation was 25 days (IQR = 0–169) and the median time span from the last LDL-C measurement to statin initiation was 6 days (IQR = 0–68). The treatment intensity of the first statin prescription was low in 2.1% (*n* = 17), moderate in 56.3% (*n* = 462) and high in 36.5% (*n* = 300); for 6% of patients, treatment intensity was missing. Table 1 describes patients stratified by CV risk and statin treatment decision. The patients’ GPs (*n* = 189) were female in 32.8% (*n* = 62) and had a median age of 51 years (IQR = 43–59).

### 3.1. Guideline Concordance of Statin Treatment Decisions

The guidelines’ recommendations were “statin initiation” in 42.3% (*n* = 3413), “statin optional” in 16.6% (*n* = 1336) and “no statin” in 41.1% (*n* = 3311) of patients. During follow up, 10.2% (*n* = 821) of patients were initiated on statins. When assessing guideline concordance, 63.0% (*n* = 5078) of treatment decisions were concordant, 35.8% (*n* = 2882) classified as undertreatment and 1.2% (*n* = 100) as overtreatment (Figure 3). Specifically, 84.4% of patients (*n* = 3413) for whom statin treatment was recommended were not initiated on a statin (undertreatment), and among patients for whom it was not recommended, it was initiated in 3.0% (overtreatment). By definition, undertreatment could only occur in the high and very high CV risk categories (for patients at high risk in 69.8%, for patients at very high risk in 75.7%) and overtreatment only in the low, moderate and high CV risk categories (for patients at low risk in 1.3%, for patient at moderate risk in 2.4%, and for patients at high risk in 0.8%). In patients for whom statin treatment was optional (*n* = 1336), it was initiated in 14.2%.

### 3.2. Determinants of Undertreatment

Patients for whom the ESC-G recommended statin initiation (*n* = 3413) were further analyzed to find determinants of undertreatment. These patients (49.0% female) were at high CV risk in 45.2% and at very high CV risk in 54.8%, had a median age of 67 years (IQR = 58–75) and a median deviation from LDL-C treatment thresholds of 1.4 mmol/L (IQR = 0.8–2.1). The patients in the undertreated subgroup (*n* = 2882) were at high risk in 46.5% and at very high risk in 53.5%, female in 49.2%, had a median age of 67 years (IQR = 58–75) and a median deviation from LDL-C treatment thresholds of 1.3 mmol/L (IQR = 0.7–2.0). Multilevel logistic regression suggested that undertreatment was associated with close proximity of LDL-C to treatment thresholds, high CV risk (compared to very high CV risk), female patient sex, low continuity of care, and high GP age (Table 2). Patients’ variables that were considered in the regression analysis are presented in Appendix A, stratified by guideline concordance of the treatment decision.

## 4. Discussion

We evaluated guideline concordance of statin treatment decisions considering LDL-C levels and CV risk for over 8000 statin-naive patients in primary care. For two-thirds of patients, the treatment decision (initiation or non-initiation) was guideline-concordant, one-third seemed undertreated and one percent overtreated. Determinants of undertreatment were: close proximity to LDL-C treatment thresholds, lower CV risk, female patient sex, low continuity of care and high GP age.

In this cohort study, we included patients without former statin treatment who had been seeing their GP for at least one year and who were under the GP’s dyslipidemia surveillance. Patients under such dyslipidemia surveillance were approximately 60 years old and half of these patients were classified to be at high or very high CV risk. Interestingly, in these patients at high or very high CV risk, statins were initiated in only approximately 15%, which according to guidelines translated to approximately 70% of undertreated patients.

A high prevalence of undertreatment with statins has been well documented in other health care systems and has been linked to different factors [9,15,16,17,18]. Our study also found several determinants of undertreatment but gives a more differentiated impression of treatment decisions: Firstly, our results indicated that undertreatment was more common in patients whose LDL-C levels were closer to the treatment thresholds. This suggests that even though recommended LDL-C treatment thresholds were not strictly adopted, LDL-C levels were indeed taken into account in the treatment decisions. In addition, we found that undertreatment was less common in patients at very high CV risk compared to patients at high CV risk and thereby were able to show that GPs account for the patients CV risk, which contrasted studies reporting that GPs placed too much emphasis on dyslipidemia rather than absolute CV risk [7,16,17]. Interestingly, among patients with very high CV risk in our study, those with established atherosclerotic CV disease (patients in secondary prevention) were more likely to be initiated on a statin than those without established atherosclerotic CV disease (primary prevention, Table 1). Thus, it is possible that even though the distinction between primary and secondary prevention has been abandoned in the guidelines (and in our study) in favor of risk categories, GPs are partly still adhering to this concept.

For both LDL-C treatment thresholds and CV risk, it is conceivable that they were interpreted as gradual transitions dependent on the individual patient, rather than strict cut offs which are identical for all patients. This would be reasonable, since medical interventions are always subject to an individual cost–benefit analysis, and the cost of the individual patients’ statin initiation versus their personally expected benefit cannot be reproduced by universal guidelines [19,20]. However, it is also possible that some of the GPs generally used other thresholds than those recommended by the current European guideline, irrespective of individual patients. Some GPs might follow the national [21] rather than the European guidelines, which have been shown, however, to closely overlap [22]. Interestingly, patients treated by older GPs were more often subject to undertreatment, which suggested that older GPs might have followed older guidelines using higher LDL-C treatment thresholds [23]. In contrast to previous studies, however, GP sex was not associated with undertreatment in our study [24]. Nevertheless, our results confirmed the well-known association between female patient sex and undertreatment. This is presumably related to both the patient and the GP, as women have been reported to decline statin treatment more frequently than men, but also to be offered statin treatment less frequently [18]. Lastly, low continuity of care was associated with undertreatment in our study which is plausible, since fragmentation of health care hampers health care delivery. For example, low continuity of primary care has also been associated with lower adherence to statins and higher LDL-C levels [25,26].

### 4.1. Strengths and Limitations

This study has the following strengths: We studied a well-defined primary care cohort comprising patients under CV risk surveillance including LDL-C measurements. For preventive medical interventions, primary care is the most important health care sector where the largest parts of the at-risk population can be identified and treated. Mindsets of GPs involved in treatment decisions are therefore crucial to public health and especially so regarding prevention of the most prevalent and preventable CV diseases. In this context, we revealed many associations with undertreatment of statins of which several can be addressed by targeted initiatives to improve quality of care and CV outcomes. Our operationalization of CV risk categories represented a faithful implementation of the ESC-G by combining the score- and the morbidities-based classification.

The first limitation to this study is that we were unable to distinguish deliberate omission of statin initiations from simple forgetfulness and therefore not all instances of undertreatment rest on deliberate decisions. Nonetheless, the determinants we found are independent from the deliberateness of the undertreatment and may still serve as targets to improve quality of care. An additional limitation is that we potentially underestimated undertreatment frequency based on the underestimation of CV risk due to missing data on comorbidities such as smoking. On the other hand, missing records of statin initiations were a potential source for overestimating the proportion of undertreatment. However, because statin dispensing requires a prescription from a physician in Switzerland, these records are rather unlikely to be missing. Cases in which GPs refrained from statin initiation because of previously diagnosed statin intolerance were a further source of overestimating the proportion of undertreatment. However, we included only statin-naive patients according to our database, and thereby minimized this potential source of undertreatment. Lastly, it must be acknowledged that treatment decisions are not solely under the influence of the GP, but also depend on the patient’s readiness to take statins as a result of their individual cost–benefit analysis, as discussed above. Prescribing statins to unconvinced patients most probably leads to low medication adherence. In fact, low adherence to statins is common, and naturally decreases their beneficial effect [27,28].

### 4.2. Implications

We found several determinants of undertreatment, such as demographic patient and GP variables, and increasing awareness of these could improve guideline concordance of statin treatment decisions. Most notably, however, we found that proximity of LDL-C levels to treatment thresholds was associated with undertreatment, indicating that when LDL-C treatment thresholds were not followed, it was partly due to the LDL-C being close to the acceptable range. This finding is meaningful in the context of the abundant critique towards GP non-adherence to dyslipidemia guidelines. GP performance is often evaluated based on cut offs, i.e., LDL-C treatment thresholds or LDL-C target values. As discussed above, however, patients’ individual cost–benefit analyses cannot be represented by guidelines. Moreover, binary evaluation according to strict cut offs does not respect that laboratory tests such as LDL-C measurements are subject to analytical and biological variation, which has been estimated at 20% for LDL-C [29]. In this study, we tried to mitigate the uncertainty of LDL-C measurements by using averages for the assessment of guideline concordance. Ultimately, it cannot be firmly established to which degree personalization of guideline recommendations is beneficial to patients, rendering judgments about appropriateness difficult.

## 5. Conclusions

Overall, the statin treatment decisions were guideline-concordant in two-thirds of patients. Even though patients at high or very high CV risk were often undertreated, we found that GPs considered CV risk and LDL-C in their treatment decisions. Guideline deviations might partly be explained by GPs following a personalized approach for each patient.

## Figures and Tables

**Figure 1 jcm-09-03719-f001:**
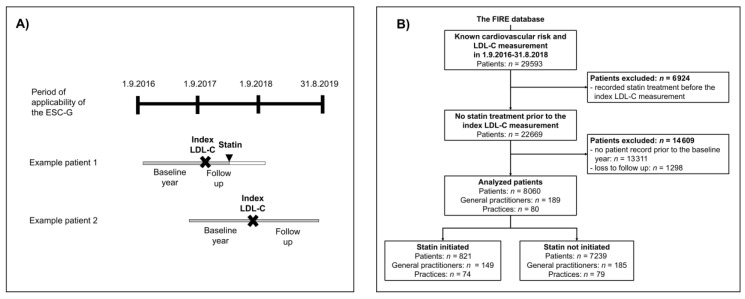
Study phases and participants. (**A**) Graphical representation of study phases for two example patients. Patient 1 was initiated on a statin; patient 2 was not initiated on a statin. The grey bar represents their individual observation period (covering baseline year and follow up). (**B**) Flowchart of the patient selection process. Abbreviations: ESC-G, 2016 European Society of Cardiology/European Atherosclerosis Society Guidelines for the management of dyslipidemias; LDL-C, low-density lipoprotein cholesterol; FIRE, Family Medicine ICPC-Research using Electronic Medical Records.

**Figure 2 jcm-09-03719-f002:**
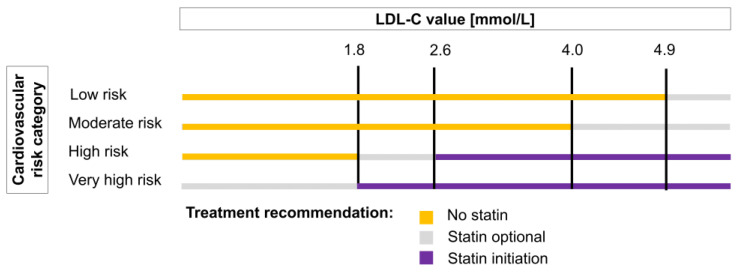
Statin treatment recommendations based on cardiovascular risk category and low-density lipoprotein cholesterol levels according to the 2016 European Society of Cardiology/European Atherosclerosis Society Guidelines for the Management of Dyslipidemias.

**Figure 3 jcm-09-03719-f003:**
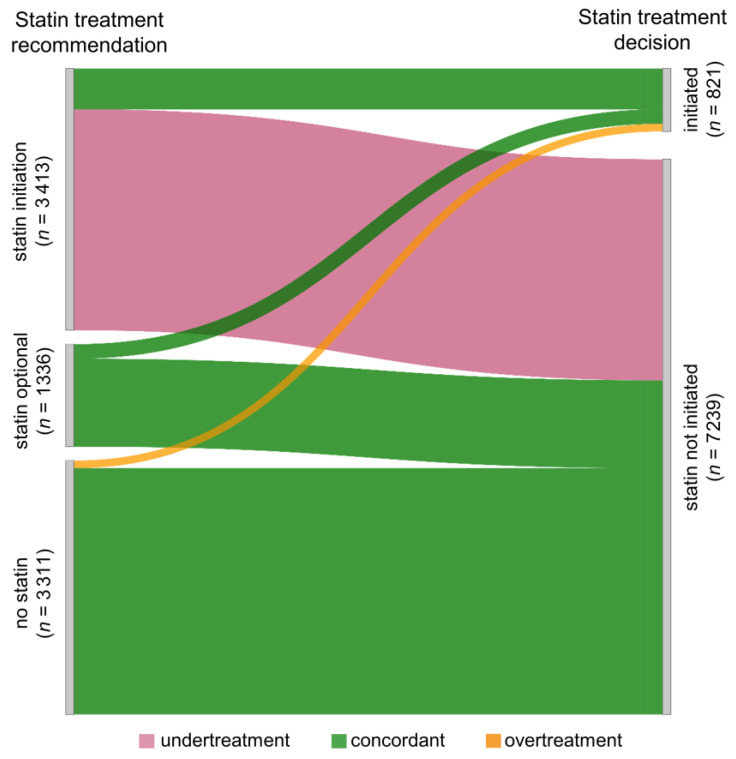
Guideline concordance of statin treatment decisions. The flows represent the patients’ statin treatment recommendations and decisions; the size of the flows is proportional to the number of patients (total *n* = 8060). The colors indicate the concordance of recommendation and treatment: green indicates guideline-concordant treatment, pink indicates undertreatment and orange indicates overtreatment.

**Table 1 jcm-09-03719-t001:** Description of patients stratified by cardiovascular risk and statin treatment decision.

Variable	Low CV Risk		Moderate CV Risk		High CV Risk		Very High CV Risk	
	Statin Initiated (*n* = 24, 2%)	Statin Not Initiated (*n* = 1160, 98%)	Statin Initiated (*n* = 152, 5%)	Statin Not Initiated (*n* = 2767, 95%)	Statin Initiated (*n* = 250, 13%)	Statin Not Initiated (*n* = 1668, 87%)	Statin Initiated (*n* = 395, 19%)	Statin Not Initiated (*n* = 1644, 81%)
Age at index	48 (45–52)	46 (42–50)	58 (54–62)	57 (53–61)	67 (61–74)	68 (60–77)	67 (58–74)	65 (55–75)
% female	75.0	79.1	37.5	41.0	56.4	52.5	39.2	44.8
LDL-C levels in mmol/L	4.5 (3.8–5.1)	3.1 (2.6–3.7)	4.2 (2.8–4.9)	3.4 (2.9–4)	4.5 (3.1–5.7)	3.5 (2.8–4.2)	3.2 (2.1–4.1)	3.2 (2.5–3.7)
Deviation from LDL-C treatment thresholds in mmol/L	-	-	-	-	1.9 (0.5–3.1)	0.9 (0.2–1.6)	1.4 (0.3–2.3)	1.4 (0.7–1.9)
COCI	1.0 (0.8–1.0)	1.0 (0.7–1.0)	1.0 (0.8–1.0)	1.0 (0.8–1.0)	1.0 (0.9–1.0)	1.0 (0.9–1.0)	1.0 (0.8–1.0)	1.0 (0.9–1.0)
Morbidities ^1^:								
% ASCVD	0.0	0.0	0.0	0.0	0.0	0.0	22.8	11.1
% diabetes	0.0	0.0	0.0	0.0	2.8	4.4	79.2	86.6
% severe CKD	0.0	0.0	0.0	0.0	0.0	0.0	5.1	5.4
% moderate CKD	0.0	0.0	0.0	0.0	46.0	62.2	21.8	20.5
% hypertension	20.8	14.7	30.3	26.3	49.2	52.9	66.8	63.1
% obesity	16.7	18.0	18.4	19.6	23.2	21.5	32.9	36.2
Statin treatment intensity:								
% low	4.2	-	2.0	-	2.0	-	2.0	-
% moderate	75.0	-	57.9	-	58.4	-	53.2	-
% high	12.5	-	34.9	-	33.6	-	40.5	-
% missing	9.0	-	6.0	-	7.0	-	5.0	-

Data are presented as the median (interquartile range) or percentages. Abbreviations: CV, cardiovascular; LDL-C, low-density lipoprotein cholesterol; COCI, continuity of care index; ASCVD, atherosclerotic cardiovascular disease; CKD, chronic kidney disease. ^1^ The definitions of morbidities in the database are given in Appendix A.

**Table 2 jcm-09-03719-t002:** Determinants of undertreatment. Patients for whom statin would be recommended were considered (*n* = 3413); 5.7% (*n* = 196) of these patients were ignored in the regression due to missing values.

Variable	OR	95% CI	*p* Value
Deviation from LDL-C treatment threshold (per decrease in 1 mmol/L)	2.09	1.87 to 2.35	<0.001
High CV risk (vs. very high)	1.64	1.30 to 2.05	<0.001
Patient female sex	1.31	1.05 to 1.64	0.018
Younger patient age (per decrease in 10 year)	1.09	0.99 to 1.20	0.068
COCI < 1 vs. COCI = 1 ^1^	1.68	1.28 to 2.19	<0.001
GP female sex	0.74	0.49 to 1.11	0.144
Younger GP age (per decrease in 10 year)	0.74	0.61 to 0.90	0.002

Abbreviations: OR, odds ratio; CI, confidence interval; CV, cardiovascular; LDL-C, low-density lipoprotein cholesterol; COCI, continuity of care index; GP, general practitioner. ^1^ Patients with a COCI = 1 were always treated by the same GP.

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
