# Peer review of "Guideline Concordance of Statin Treatment Decisions: A Retrospective Cohort Study"

_jcm, 2020, doi:10.3390/jcm9113719_

Round 1

Reviewer 1 Report

The authors conducted a retrospective analysis of electronic medical records of statin naive patients to see how well primary care physicians follow recommendations for lipid lowering therapy.  The study is interesting, and the manuscript well written.  The conclusion is that a significant number of patients who are eligible for statin therapy are undertreated and that many physicians do not follow guideline based recommendations for lipid lowering therapy.  However, a significant limitation of this study is that patients were excluded if they had been treated with statins at or prior to the index date at which the LDL cholesterol was measured.  This introduces some bias into the study in that a population of patients may have been selected who were statin intolerant.  It would be helpful if information was available to see if any of these patients had been on a statin in the past which would suggest intolerance.  Also is there any information from the database regarding statin intolerance?

Specific points:

  1. In the method section figure 1B notes that a number of patients were excluded because of insufficient baseline data.  The authors need to define what baseline data was necessary to be enrolled in the study.  For example were these patients excluded because their cardiovascular risk category could not be defined?
  2. I am also surprised by the finding that 81% of “very high cardiovascular risk” patients were not prescribed a statin. The 2016 ESC/EAS cholesterol guidelines included both patients with ASCVD and very high risk primary prevention patients in this category.  It would be interesting to sort this out.  What % of patients with clinical ASCVD were prescribed a statin verses those who were very high risk primary prevention patients.  It may well be that most of the under-treatment in this group was in the primary prevention group.

Author Response

We thank the Reviewer 1 for the valuable comments.

Reviewer 1 suggests that the exclusion of patients who received statins prior to the LDL-C measurement index date introduces some bias into the study in that a population of patients may have been selected who were statin intolerant. 

We agree with Reviewer 1 that it is a limitation of the study that we did not have information on statin intolerance, and added this limitation to the according section (page 7 line 247): “A further source of undertreatment are cases in which GPs refrained from statin initiation because of previously diagnosed statin intolerance. However, we included only statin-naive patients according to our database, and thereby minimized this potential source of untertreatment.”

Regarding the specific points raised by Reviewer 1:

  1. Reviewer 1 rightly points out that it is unclear what was meant by “baseline data”. In fact, “baseline data” was simply defined as a patient record prior to the baseline year (index year – 1 year) to make sure that the patients had records in the database for a sufficient period, in order to know their medication, morbidities, etc. We adapted Figure 1B accordingly.
  2. As Reviewer 1 rightly suspects, within the very high risk group, those without ASCVD (representing the primary prevention group) were less likely to be prescribed a statin compared to those with ASCVD (no statin in 66.9% vs 82.7%). Thus, it is possible that even though the distinction between primary and secondary prevention has been abandoned in favor of risk categories, GPs are partly still adhering to this concept. We added an according comment to page 7 line 205.

Reviewer 2 Report

Article is interesting. The study uses a proper methodology. Introduction, Experimental Section, Results and Discussion sections have been properly done.

Minor:

No information about the method of LDL-C measurements can be found. Was Friedewald formula used to calulate LDL-C or direct method was done?

Author Response

We thank the Reviewer 2 for this positive assessment.

Regarding the LDL-C values: the LDL-C methodology depends on the general practice and is missing in the database.

Since the ESC guidelines apply to both direct and calculated (Friedewald formula) LDL-C measurement methods and reportedly do not differ relevantly (Mora et al., Clin Chem. 2009 May; 55(5): 888–894. doi: 10.1373/clinchem.2008.117929) this aspect was disregarded in our study. We refrained from adding this information to the manuscript.

Reviewer 3 Report

The study by Rachamin et al. provides a detailed focus on the behaviour of statin prescribers in primary setting in an European cohort, suggesting some ideas to further improve CV risk management at the population level. The methodology is rigorous and results reliable. The manuscript is weel-written.

I would suggest the Authors to add one additional comment. Table 1 clearly shows that moderate intensity statin treatment is the preferred one across all CV risk subgroups (low-moderate-high-very high). Could also this finding provide a measure of statin undertreatment in patients at high and very high CV risk or not? Indeed, high intensity statin treatment could be expected to be the preferred one in patients at high and very high CV risk. Undoubtely, it was far from the study objectives to analyze this aspect. Nonetheless, from a public health perspective the choice of statin treatment intensity may matter, beyond the choice of statin treatment.

Author Response

We thank the Reviewer 3 for this positive feedback. The Reviewer 3 raises a relevant point, i.e. the contribution of statin treatment intensity (as opposed to the binary choice of treatment) to undertreatment. However, we would like to point out that the intensities reported in our study are those of the first statin prescription, and it can be assumed that GPs often start statins at a moderate dose before increasing the intensity. Perhaps Reviewer 3 would enjoy our previous publication that discussed the remaining potential to increase the intensity of statin treatment in order to improve LDL-C target achievement (Meier et al., J. Clin. Med. 2020, 9, 2140; doi:10.3390/jcm9072140)